

# LINflow: a computational pipeline that combines an alignment-free with an alignment-based method to accelerate generation of similarity matrices for prokaryotic genomes

Long Tian[1,*], Reza Mazloom[2,*], Lenwood S. Heath[2] and Boris A. Vinatzer[1]

[1] School of Plant and Environmental Sciences, Virginia Tech, Blacksburg, VA, USA
[2] Department of Computer Science, Virginia Tech, Blacksburg, VA, USA
* These authors contributed equally to this work.

Corresponding author
Boris A. Vinatzer, vinatzer@vt.edu

## ABSTRACT

**Background:** Computing genomic similarity between strains is a prerequisite for genome-based prokaryotic classification and identification. Genomic similarity was first computed as Average Nucleotide Identity (ANI) values based on the alignment of genomic fragments. Since this is computationally expensive, faster and computationally cheaper alignment-free methods have been developed to estimate ANI. However, these methods do not reach the level of accuracy of alignment-based methods.

**Methods:** Here we introduce LINflow, a computational pipeline that infers pairwise genomic similarity in a set of genomes. LINflow takes advantage of the speed of the alignment-free sourmash tool to identify the genome in a dataset that is most similar to a query genome and the precision of the alignment-based pyani software to precisely compute ANI between the query genome and the most similar genome identified by sourmash. This is repeated for each new genome that is added to a dataset. The sequentially computed ANI values are stored as Life Identification Numbers (LINs), which are then used to infer all other pairwise ANI values in the set. We tested LINflow on four sets, 484 genomes in total, and compared the needed time and the generated similarity matrices with other tools.

**Results:** LINflow is up to 150 times faster than pyani and pairwise ANI values generated by LINflow are highly correlated with those computed by pyani. However, because LINflow infers most pairwise ANI values instead of computing them directly, ANI values occasionally depart from the ANI values computed by pyani. In conclusion, LINflow is a fast and memory-efficient pipeline to infer similarity among a large set of prokaryotic genomes. Its ability to quickly add new genome sequences to an already computed similarity matrix makes LINflow particularly useful for projects when new genome sequences need to be regularly added to an existing dataset.

# INTRODUCTION

The number of prokaryotic genome assemblies available at the National Center for Biotechnology Institute (NCBI) is growing rapidly and has reached 615,000 in 2020. It can be anticipated that many more genome assemblies will be published in the near future because of continued improvements in next-generation DNA sequencing technologies concerning throughput and sequence quality and a concomitant drop in sequencing cost. The ever-growing collection of prokaryotic genomes provides the opportunity to explore evolutionary relationships among species, genomic boundaries of species and the genetic diversity within species.

DNA–DNA hybridization (DDH) was the first method that incorporated genome content in prokaryotic classification and became the gold standard in the 1970s (*Brenner, 1973*). Two strains that have a reciprocal DDH value of over 70% are considered to belong to the same species (*Brenner, 1973*). However, the low resolution, laborious experimental procedures, and limited portability of results present serious limitations (*Stackebrandt & Goebel, 1994*). After the advent of next-generation sequencing, DDH has largely been replaced by Average Nucleotide Identity (ANI). ANI is a measure of similarity between two genomes based on the comparison of whole genome sequences (*Konstantinidis & Tiedje, 2005a*). In its original implementation, one genome is used as a query genome and is cut into consecutive 1,020 nt-long fragments. Each fragment is then aligned to the second genome, the subject genome, using BLAST (*Konstantinidis & Tiedje, 2005a*). Alignments that result in over 30% coverage and 70% identity are retained and ANI is computed as the average identity of these alignments. An ANI between 95% and 96% has been found to correspond to 70% DDH (*Goris et al., 2007*). ANI even provides the resolution necessary to separate strains into different genome similarity groups within species (*Rodriguez-R et al., 2018*; *Vinatzer et al., 2016*).

While computing ANI between a single pair of genome sequences using the alignment-based BLAST algorithm is reasonably efficient, computing all pairwise ANI values for thousands of genomes (thus requiring up to millions of pairwise comparisons) is slow. Therefore, various methods have been developed to infer ANI based on alignment-free algorithms. In 2015, Ondov and colleagues published the first implementation of MinHash (*Broder, 1997*) for prokaryotic genome comparisons, Mash (*Ondov et al., 2016*). Mash and similar tools, such as sourmash (*Brown & Irber, 2016*; *Pierce et al., 2019*), produce a reduced representation of a genome as a sketch (also referred to as a signature). This is done by determining the presence of all k-mers in a genome sequence and using a hash function to translate these k-mers into hashes, of which a subset is used as the sketch. Mash and sourmash then compare genomes by calculating the Jaccard similarity between their sketches (*Ondov et al., 2016*). This results in an estimate of the Jaccard similarity between the entire k-mer sets of the two genomes. Not only was it possible with this approach to process

and calculate the pairwise similarity of 54,118 microbial genomes from NCBI RefSeq release 70 in 33 CPU hours, but this approximate Jaccard similarity also correlates with ANI almost linearly for ANI values from approximately 90% to 99%. Therefore, Mash and sourmash can be used to precisely and quickly cluster genomes into species.

The $k$ value, that is, the k-mer length employed when a sketch is made, significantly impacts the computed Jaccard similarity. A smaller $k$ value enables the detection of similarity between genomes of distantly related strains but loses resolution when the ANI between genomes is high. On the other hand, longer k-mers provide high resolution when ANI is high but they cannot detect any similarity between genomes of more distantly related strains (*Brown & Irber, 2016*; *Pierce et al., 2019*).

FastANI is another tool to determine how similar genome sequences are to each other (*Jain et al., 2018b*, *2018c*). However, instead of building a sketch of a whole-genome sequence, FastANI maintains the conceptual framework of BLAST-based ANI: it breaks the query genome into non-overlapping fragments and only in the next step replaces BLAST with an alignment-free k-mer approach, called MashMAP (*Jain et al., 2018a*). FastANI is 50–4,000 times faster than BLAST-based ANI, while inferring ANI accurately for ANI values as low as 80% (*Jain et al., 2018b*, *2018c*).

The Life Identification Number (LIN) system is a genome similarity-based system to classify individual organisms based on reciprocal ANI (*Marakeby et al., 2014*; *Vinatzer, Tian & Heath, 2017*; *Vinatzer et al., 2016*; *Weisberg et al., 2015*). A LIN consists of a series of positions, where each position indicates an ANI threshold, from low to high, starting from the leftmost position. The LIN of a genome is assigned based on the ANI to its most similar genome whose LIN has been already assigned. Therefore, the more similar two genomes are to each other, the longer their LINs are identical starting from the leftmost position. A group of strains sharing the same leading part of LINs is called a LINgroup, denoted by the shared part of their LINs. It has been shown that LINgroups can be used to circumscribe groups of prokaryotes from the genus level to the intraspecies level, almost reaching outbreak resolution (*Vinatzer et al., 2016*).

To analyze the diversity of a collection of prokaryotic genomes, computing all pairwise comparisons cannot be avoided by any of the above methods and their implementations. However, when dealing with a large number of genomes, pairwise comparisons are computationally expensive. Furthermore, because of the ever-growing number of sequenced genomes and their frequent addition to existing datasets, re-analysis of datasets each time new genomes are added becomes necessary.

Here we alleviate the bottleneck of pairwise ANI computations by developing LINflow, a pipeline that efficiently constructs highly resolved similarity matrices from 70% to 99.9% ANI by combining the speed of the MinHash-based sourmash tool (*Brown & Irber, 2016*; *Pierce et al., 2019*) with the precision of the BLAST-based pyani tool (*Pritchard, 2014*) and the LIN concept (*Weisberg et al., 2015*). The obtained results can then provide the basis for genome-based classification of prokaryotes.

## METHODS

### Overview

In short, to minimize the number of computationally expensive ANI computations when constructing a genomic similarity matrix, LINflow sequentially adds genomes to a dataset, at each step efficiently identifies the genome already in the dataset that is most similar to the newly added genome, precisely calculates the ANI value only between this pair of genomes, and assigns a LIN to the new genome based on this ANI value and the LIN of the most similar genome. The LINs are then used to further accelerate the identification of the most similar genome and, most importantly, to efficiently infer all remaining pairwise ANI values to construct the complete genome similarity matrix. In other words, the main purpose of LINs in LINflow is to reduce the number of necessary ANI computations to one per genome when computing a genome similarity matrix.

The LINflow flowchart is shown in Fig. 1. When a new genome is added to the dataset, LINflow identifies the genome that is most similar to this new genome among the genomes that were previously added using the computationally efficient alignment-free tool sourmash (*Brown & Irber, 2016*; *Pierce et al., 2019*). This is accomplished via a two-step procedure, which consists of first identifying the LINgroup to which the new genome belongs and then identifying the most similar genome in this LINgroup. By default, LINflow uses the 95%-level LINgroup in this step, but this can be modified by the user based on the expected genomic similarities in a specific dataset. The precise ANI value between the two genomes is then computed using the more computationally expensive, but precise, pyani tool (*Pritchard, 2014*). LINflow uses this ANI value to assign a LIN to the new genome based on the LIN of its most similar genome (which LIN was previously assigned based on the ANI value to its most similar genome when that genome itself was previously added to the dataset). The assigned LINs can then be used to infer all-against-all ANI values even though only a single ANI computation is performed for each genome.

To make the results reusable and easily accessible in terms of reading and writing, a relational database managed by MySQL (https://dev.mysql.com/) is used to store data, with the schema shown in Fig. 2. This relational database connects tables with primary and foreign keys, and the connections between tables are represented by arrows. The genome table stores the locations of the genome sequences. The taxonomy table stores the taxonomic information corresponding to each genome in the database. LIN schemes (i.e., the number of LIN positions and the corresponding ANI thresholds), based on which LINs are assigned, are kept in the Scheme table. Besides three default LIN schemes, new schemes can be added by users so that LINs can be assigned according to the users' needs in resolution. LINs are assigned with the three default schemes and with one user-defined scheme if there is any.

The individual steps of the pipeline are described in detail below.

### Generation and storing of signatures

LINflow uses sourmash version 2.0 (*Pierce et al., 2019*) to generate signatures for both $k = 21$ and $k = 51$ with $n = 2,000$ (i.e., each signature consists of 2,000 hashes) for all genomes and stores them as individual files. The first member of each 95%-level LINgroup

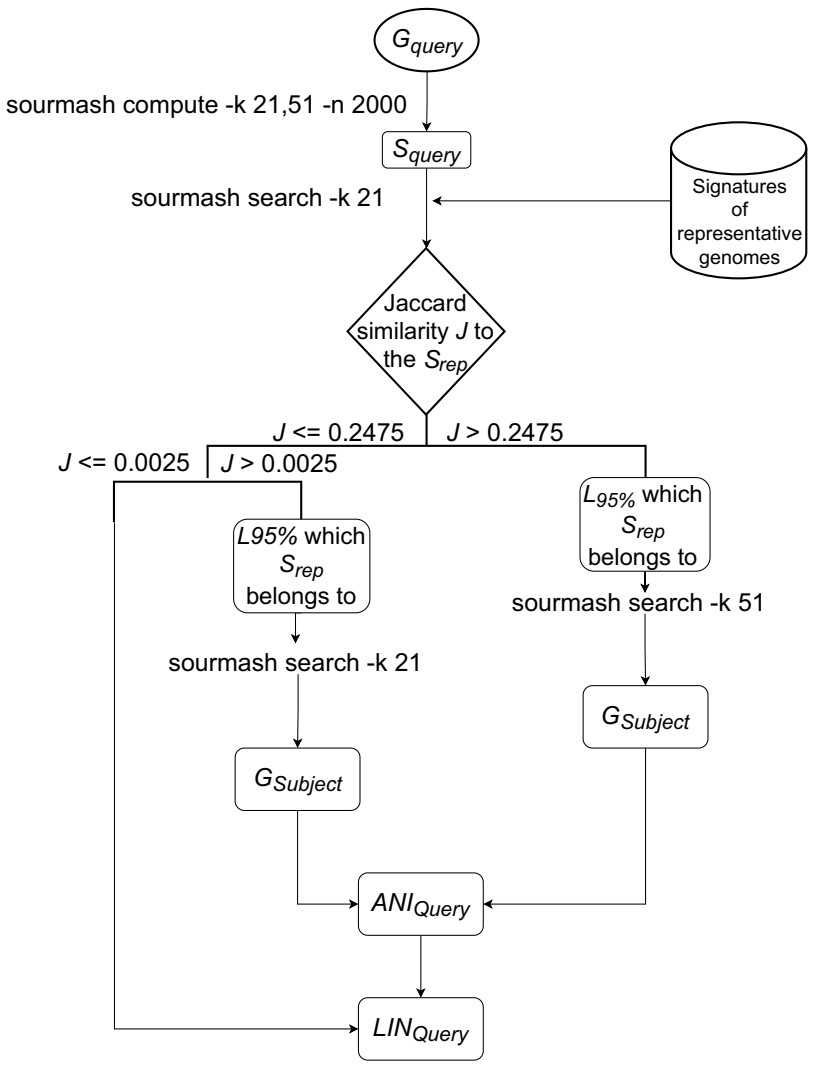

**Figure 1  Workflow of LINflow.** The flowchart of the LIN assignment algorithm used in LINflow.

is chosen as the representative genome and a copy of its signature file is saved in a second directory together with signature files of all other representative genomes.

## Choice of LIN scheme

LINflow allows the user to choose from four default LIN schemes: (1) a 20-position LIN scheme that ranges from 70% ANI to 99.999% ANI to cope with genus to strain level differentiation and is currently used in LINbase (*Tian et al., 2020*) (Table 1), (2) a 300-position scheme with positions starting at 70% and increasing by 0.1, and reaching 99.9% (used in this manuscript with the datasets listed below), (3) a 3,000-position scheme starting at 70% and reaching 99.99% at 0.01% intervals (recommended for constructing highly resolved similarity matrices for strains belonging to the same species), (4) a 300,000-position scheme starting at 70% and reaching 99.99999% at 0.00001% intervals. The user can also define any custom LIN scheme with any number of
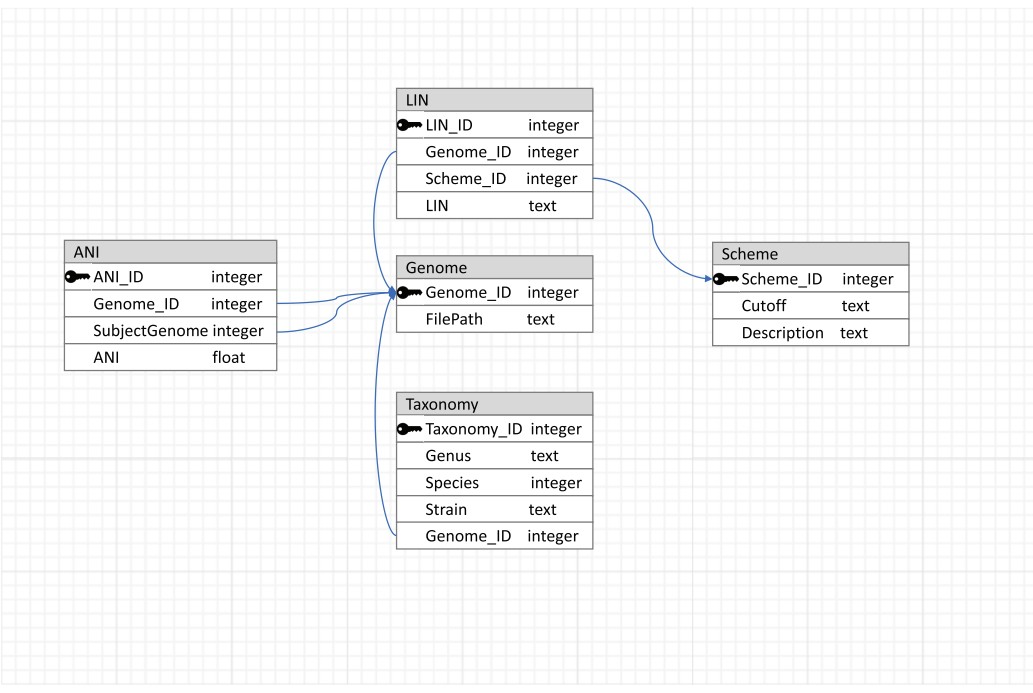

**Figure 2 Database schema used by LINflow.** The relational database connects tables with primary and foreign keys. Connections between tables are represented by arrows.

**Table 1 The primary LIN assignment scheme of LINbase used to assign LINs by LINflow in this study.**

| ANI | 70 | 75 | 80 | 85 | 90 | 95 | 96 | 97 | 98 | 98.5 | 99 | 99.25 | 99.5 | 99.75 | 99.9 | 99.925 | 99.95 | 99.975 | 99.99 | 99.999 |
|---|---|---|---|---|---|---|---|---|---|---|---|---|---|---|---|---|---|---|---|---|
| **Position** | A | B | C | D | E | F | G | H | I | J | K | L | M | N | O | P | Q | R | S | T |

positions using any ANI value of choice (however, it is not advised to use ANI values below 70% since ANI does not reflect evolutionary relationships when ANI falls below 70%).

## Initiation of LINflow

LINflow, by default, includes the 20-position LIN scheme as one of the schemes during each run. An arbitrary genome will be selected to assign the first LIN with 0 in each position. Its signature will be generated and saved as the representative of the LINgroup $0_A0_B0_C0_D0_E0_F$ using the 20-position default scheme both in the directories for representative genomes and this LINgroup.

## LIN assignment

The new genome's ($G_{Query}$) signature $S_{Query}$ will be first queried against the representative genomes of all existing 95%-level LINgroups (or F LINgroups) with $k = 21$, using sourmash. Based on the analysis of more than 6,000 bacterial genomes from different families with $k = 21$, the Jaccard similarity of 0.2475 was found to correspond to 95% ANI (data not shown). If the highest Jaccard similarity $J$ to one of the representative genomes

$S_{\text{Query}}$ is above 0.2475, then the corresponding genome represents the 95%-level LINgroup ($L_{95\%}$), belongs to. If $0.0025 < J <= 0.2475$, then the corresponding genome is at least 70% similar to $G_{\text{Query}}$, which means they share at least the A position in the default LIN scheme. If $J <= 0.0025$, the corresponding genome is less than 70% similar to $G_{\text{Query}}$.

If $J > 0.2475$, $S_{\text{Query}}$ will be queried against all members of $L_{95\%}$ by sourmash with $k = 51$. The most similar genome $G_{\text{Subject}}$ according to Jaccard similarity in $L_{95\%}$ is identified as the most similar genome to $G_{\text{Query}}$ in the entire database.

If $0.0025 < J \leq 0.2475$, $S_{\text{Query}}$ will be queried against all the members of $L_{95\%}$ by sourmash with $k = 21$. The most similar genome $G_{\text{Subject}}$ according to Jaccard similarity in $L_{95\%}$ is identified as the most similar genome to $G_{\text{Query}}$ in the entire database.

For the above cases, ANI between $G_{\text{Query}}$, and $G_{\text{Subject}}$, $\text{ANI}_{\text{Query}}$, will be calculated with pyani. To assign the LIN to the query genome, $G_{\text{Subject}}$'s LIN, $\text{LIN}_{\text{Subject}}$, will be used as the reference from A to the last position that the ANI threshold is lower than or equal to $\text{ANI}_{\text{Query}}$, the first position that ANI threshold is larger than $\text{ANI}_{\text{Query}}$ will be assigned a number that has not been used with the prefix in the database, and the rest of the positions will be filled with zeros. For example, $\text{ANI}_{\text{Query}} = 95.4575\%$, it is over 95% at the F position but lower than 96% at the G position, so it will use $\text{LIN}_{\text{Subject}}$ from A to F as $\text{LIN}_{\text{Query}}$'s A to F. At $\text{LIN}_{\text{Query}}$'s G position, a number that has never been used together with $\text{LIN}_{\text{Subject}}$'s prefix from A to F will be assigned. Each of $\text{LIN}_{\text{Query}}$'s H to T positions will be assigned 0.

If $J \leq 0.0025$, no genome in the current database has over 70% ANI to $G_{\text{Query}}$, so that a new number that has never been used in the A position before will be assigned to $\text{LIN}_{\text{Query}}$'s A position, and the rest of $\text{LIN}_{\text{Query}}$ will be filled with zeros.

### Update of database and signature file system

$G_{\text{Query}}$, $G_{\text{Subject}}$, $\text{ANI}_{\text{Query}}$ and $\text{LIN}_{\text{Query}}$ will all be written to the database. If $\text{LIN}_{\text{Query}}$ creates a new 95%-level LINgroup, a new directory for this LINgroup will be created and $S_{\text{Query}}$ will be saved in this directory as a member and as a representative genome with other representative genomes, otherwise, $S_{\text{Query}}$ will be only saved in the existing 95%-level LINgroup it belongs to.

### Datasets

We compared the performance of LINflow with sourmash, pyani, and FastANI for four datasets of 484 genomes in total. Dataset A includes 248 genome sequences belonging to the genus *Pseudomonas*. Among the 248 genomes, 222 are from 46 named species. The remaining 26 genomes are from yet unnamed and undescribed *Pseudomonas* species. Dataset B consists of 43 *Xanthomonas perforans* genomes. Dataset C includes 140 genomes of the genus *Lysinibacillus*, whereby 96 of them belong to 27 named species. Dataset D includes 53 genomes from the species *Xylella fastidiosa* and two genomes from the species *Xylella taiwanensis*. A separate dataset E with the genome sequence of *Pseudomonas caeni* DSM 24390 was used to assess the computational speed of the above tools when adding a new genome to the already-analyzed dataset A. Table S1 lists the genomes included in each dataset and the actual genome sequences can be accessed directly in this repository: https://code.vt.edu/linbaseproject/linflow_datasets.

**Table 2 Software, sub-commands and parameters used for pyani, sourmash, FastANI and LINflow.**

| Software | Version | Parameters |
|---|---|---|
| Pyani[1] | 0.2.9 | -m ANIb -worker 100 |
| sourmash | 2.0.0 | -k 21, 51 -n 2000 |
| FastANI | 1.2 | -k 16 -t 1 |

**Note:**
[1] Multiprocessing was enabled.

## Comparison of tools

LINflow was compared with pyani (blast option), sourmash and FastANI in regard to speed, memory usage and accuracy. Parameters used when running these programs are listed in Table 2. Note that for each pair of genomes A and B, pyani computes ANI by calculating both the ANI of A to B and the ANI of B to A. We used the average of the two pairwise ANI values. The calculations were executed on a 2.4 GHz CPU on Cascades, an Advanced Research Computing system at Virginia Tech and the execution time and memory usage were monitored by the job scheduler built in the system.

Hierarchical clustering using the complete linkage method was applied to the similarity matrices of each dataset calculated by the four software suites using custom R scripts. Heatmaps were generated based on the hierarchical clustering results to investigate whether FastANI, LINflow and sourmash are able to classify bacteria into species as accurate as pyani. Rows and columns of the heatmaps were reordered to be in the same order as the heatmap generated by pyani.

Finally, pairwise Mantel tests (*Mantel, 1967*) in combination with Pearson correlation coefficients were computed using custom R scripts to determine how well the similarity matrices (ANI for LINflow, FastANI and pyani and Jaccard similarity for sourmash) obtained with the different tools correlated with each other.

## RESULTS

### Computational speed and memory usage

The CPU time needed to analyze datasets A, B, C and D by each software is shown in Table 3. For sourmash, two separate times are indicated since sourmash runs two separate commands: "sourmash compute", which computes signatures and "sourmash compare", which computes the pairwise Jaccard similarity between signatures. Although the FastANI workflow is also split into two phases, the indexing phase and the compute phase, the two phases cannot be executed separately. Therefore, only the total execution time is shown.

Sourmash was the fastest out of the four tools, being 1,000–4,000 times faster than pyani, depending on the dataset. FastANI and LINflow took a similar time to execute compared to pyani, being 84–253 times and 20–150 times faster, respectively.

The memory usage of LINflow was lower than pyani and lower than FastANI when analyzing datasets A and C, which have relatively larger numbers of genomes compared to datasets B and D (Table 4). Furthermore, the time cost for adding a new genome to an existing dataset by LINflow did not increase as significantly as for pyani and FastANI.

**Table 3 Runtime of each software and sub-command used to analyze data sets A, B, C and D.**

| Data set | No. of genomes | pyani[1] | sourmash compute | sourmash compare | FastANI[1] | LINflow |
|---|---|---|---|---|---|---|
| A | 247 | 60,862 min 6 s | 15 min 3 s | 14 s | 401 min 18 s | 829 min 39 s |
| B | 43 | 2,107 min 18 s | 2 min 34 s | 2 s | 17 min 49 s | 105 min 50 s |
| C | 140 | 7,274 min 18 s | 7 min 43 s | 8 s | 86 min 12 s | 171 min 38 s |
| D | 55 | 3,292 min 37 s | 1 min 41 s | 3 s | 13 min 25 s | 22 min 58 s |

**Note:**
[1] Total CPU time is listed for pyani and FastANI although multiprocessing was enabled.

**Table 4 Memory usage (GB) of each software and sub-command used to analyze data sets A, B, C and D.**

| Data set | pyani | sourmash compute | sourmash compare | FastANI | LINflow |
|---|---|---|---|---|---|
| A | 5.8 | 0.05 | 1.4 | 4.6 | 0.6 |
| B | 1.2 | 0.04 | 0.1 | 0.5 | 0.3 |
| C | 5.7 | 0.05 | 1.2 | 2.2 | 0.3 |
| D | 1.3 | 0.05 | 0.1 | 0.4 | 0.3 |

**Table 5 Runtime of each software and sub-command used to add a single new genome to the analyzed data set A.**

| Software and sub-command | CPU time |
|---|---|
| pyani | 258 min 23 s |
| sourmash compute | 0 min 4 s |
| sourmash search | 0 min 11 s |
| FastANI | 4 min 4 s |
| LINflow | 3 min 36 s |

This can be seen by comparing the time cost for adding a new genome to dataset A (Table 5) with the average processing time for each genome in dataset A (Table 3).

## Accuracy

Similarity matrices obtained with sourmash, FastANI and LINflow were compared to those obtained with pyani, which we considered the gold standard, since it is exclusively based on BLAST reflecting the original description of ANI (*Konstantinidis & Tiedje, 2005a, 2005b*). For sourmash, we determined its performance separately for $k = 21$ and $k = 51$. For LINflow, we used two of the four default schemes: the 20-position scheme used in LINbase (*Tian et al., 2020*) (see Table 1 for the LIN scheme and Table S1 for the result) and the 300-position scheme (with ANI values increasing from 70% at the left-most LIN position to 99.9% at the right-most LIN position with 0.1% intervals between neighboring positions). The LINbase scheme was used to assign LINs to the genomes and classify them as LINgroups. The 300-position scheme was used to determine the ANI similarity matrix. After similarity matrices were computed with all tools, heatmaps were generated to visualize the genomic relatedness among the analyzed genomes.

Heatmaps derived from the ANI matrices obtained with pyani (Fig. 3A), FastANI (Fig. 3B) and LINflow (Fig. 3C) show the same species level (ANI ≥ 95%) clustering of the 247 *Pseudomonas* genomes of dataset A visible as red blocks along the diagonal. Five major clusters are easily visible. Cluster 1 consists of genomes belonging to *P. aeruginosa*, cluster 2 represents the species *P. chlororaphis*, clusters 3, 4 and 5 constitute the *P. syringae* species complex and related genomes. Note that LINflow not only classified the genomes as species but also distinguished intraspecific groups as LINgroups. The LIN prefixes denote LINgroups and show both intergroup and intragroup relationships.

Sourmash was able to perform species-level clustering with $k$ values of both 21 (Fig. 3D) and 51 (Fig. 3E). The obtained results suggest that Jaccard similarity calculated with $k = 51$ only weakly correlates with ANI for low ANI values compared to $k = 21$, for example, clusters 2, 4 and 5. Instead of genomes that are highly similar to each other, for example, genomes in cluster 5, $k = 51$ provides higher resolution than $k = 21$.

To further evaluate the accuracy of LINflow compared to the other tools, the complete similarity matrices obtained for all four datasets with all tools were compared with each other using the Mantel test (*Mantel, 1967*) using Pearson's correlation coefficients. Results are reported in Fig. 4. One can easily see how the results obtained with LINflow are highly correlated with those obtained with pyani for datasets A, C and D (Pearson correlation coefficient of 0.99 or 1) but not for dataset B, which has a Pearson correlation coefficient of only 0.78. Since many pairs of genomes in set B have ANI values above 99.8% and differ from each other by less than 0.1%, we computed ANI values for set B also using the 3,000-position and a 300,000-position LIN scheme hypothesizing that the lower correlation was due to rounding of ANI values to the first decimal place when using the 300-position LIN scheme. However, switching to the 3,000-position and 300,000-position LIN scheme only increased the correlation between LINflow and pyani slightly to a Pearson correlation coefficient of 0.82.

When comparing Pearson Correlation coefficients for the FastANI vs. pyani comparison and the sourmash vs. pyani comparison with the LINflow vs. pyani comparison, LINflow shows the same or higher correlation with pyani for all datasets with the exception of dataset B, for which LINflow shows the lowest correlation with pyani.

Finally, to determine how LINflow and FastANI correlate with pyani from low to high pairwise ANI values, we plotted all pairwise ANI results obtained with LINflow and FastANI against pyani results similar to what was done by *Jain et al. (2018c)*. Figure 5 shows how FastANI and LINflow both correlate very well with pyani for ANI values above 85% but deviate from pyani at ANI values from 85% to 70%. While ANI values computed with FastANI are higher than the corresponding pyani ANI values in this range, ANI values inferred by LINflow merge into a relatively small number of ANI values for many different pyani ANI values (see "Discussion" for an explanation of this phenomenon).

## DISCUSSION

Here we developed a new tool, LINflow, to efficiently compute genome similarity matrices for genome-based classification of prokaryotes. We compared the performance of LINflow

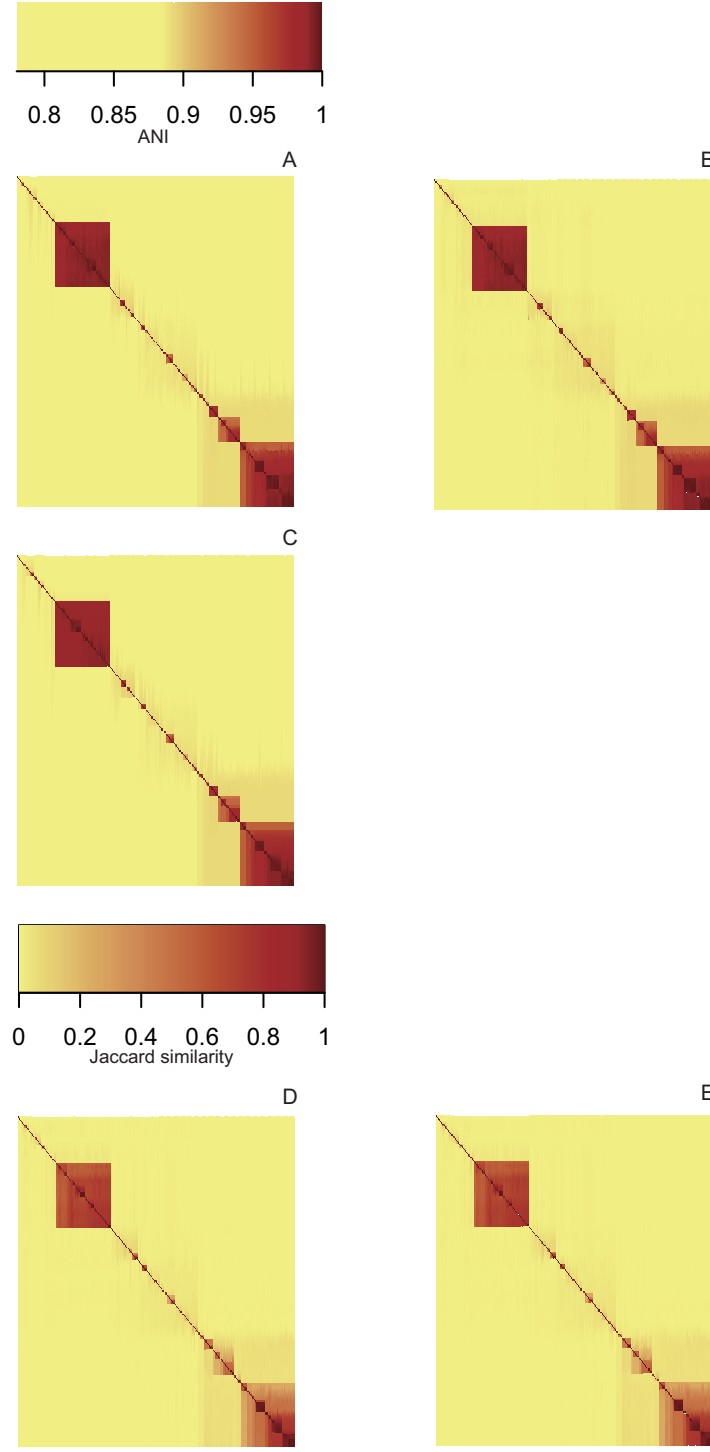

**Figure 3 Similarity matrices obtained by LINflow, FastANI, sourmash and pyani.** Heatmaps based on hierarchical clustering using the complete linkage method using the similarity matrices obtained with pyani (A), FastANI (B), LINflow (C), sourmash $k = 21$ (D) and $k = 51$ (E) for dataset A. Cluster 1 corresponds to *P. aeruginosa*, cluster 2 represents *P. chlororaphis*, clusters 3, 4 and 5 are different phylogroups within the *P. syringae* species complex and related genomes. The same figure showing strain names is included as Figs. S1–S5 (corresponding to (A) through (E)). See Figs. S6–S10, S11–S15 and S16–S20 for heatmaps of datasets B, C and D.

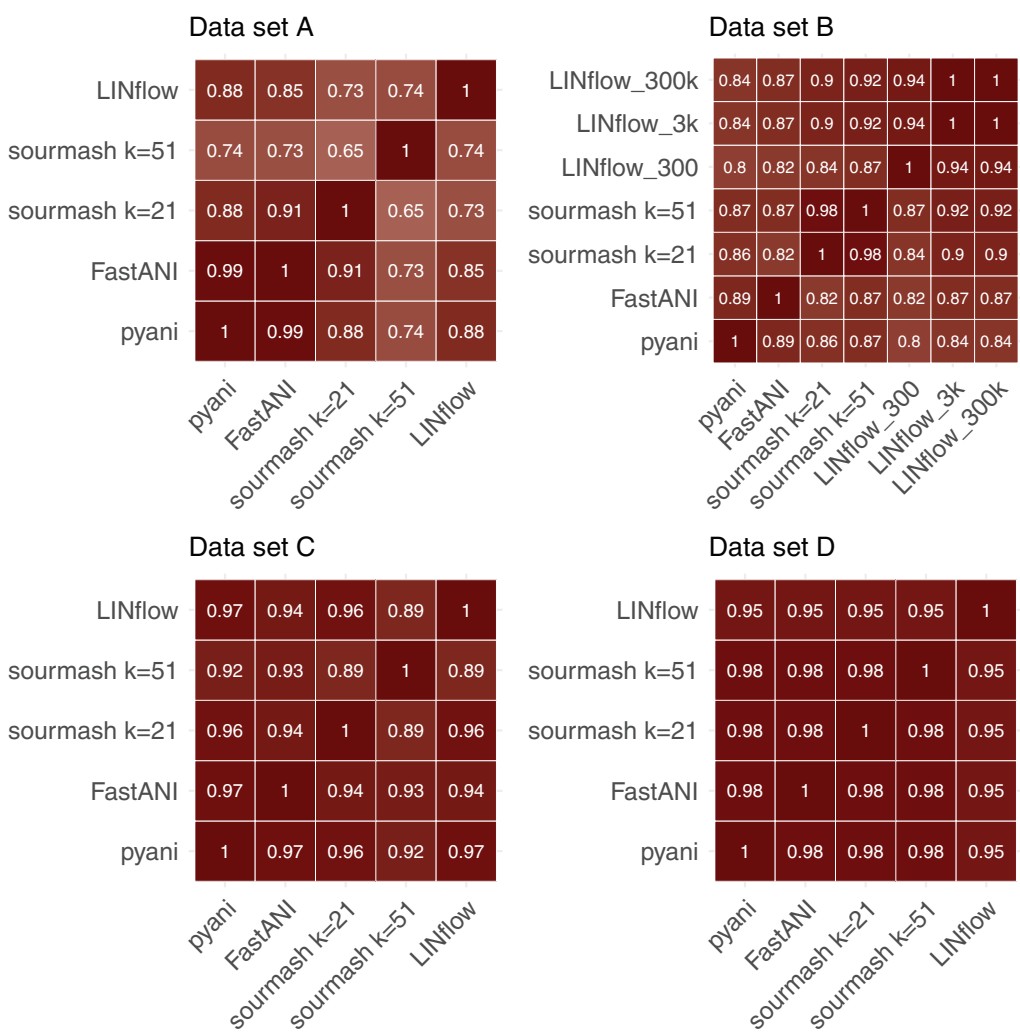

**Figure 4 Comparison of similarity matrices obtained by LINflow, FastANI, sourmash and pyani.** Heatmaps showing Pearson correlation coefficients based on the Mantel test performed between the similarity matrices obtained with pyani, sourmash, FastANI, and LINflow for datasets A, B, C and D.

with that of sourmash, FastANI and pyani when analyzing four sets of genomes and when adding a new genome to an already-analyzed dataset.

The execution time to compare each new genome to a growing dataset does not increase as significantly for LINflow as for the other tools. This is because, for each genome, the LINflow algorithm involves only a one-time sourmash signature generation, at most two sourmash signature comparisons and a one-time two-way ANI calculation with pyani between the new genome and its most similar genome identified by sourmash. We thus expect that, with larger datasets, LINflow will outperform FastANI in terms of speed and memory usage and that the relative increase in speed compared with pyani will be even more significant.

The LIN approach had been shown previously to correlate well with core genome phylogenetic trees within the genus *Pseudomonas* (*Vinatzer et al., 2016*). Previous

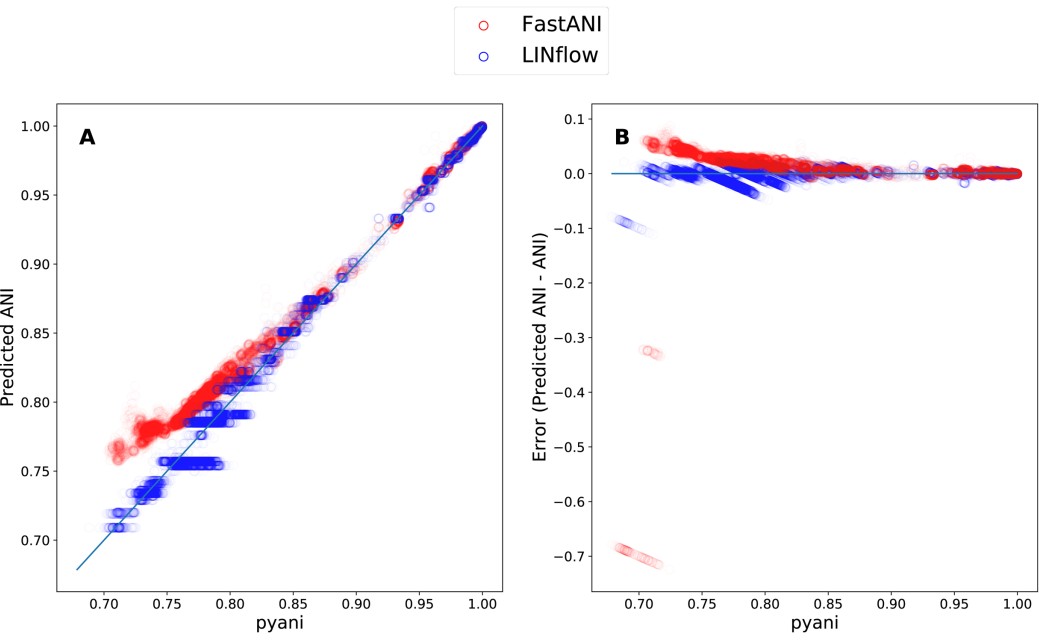

**Figure 5 Correlation between the ANI results obtained with LINflow and FastANI and the ANI results obtained with pyani (using the BLAST option) for datasets A through D.** (A) Plot showing the ANI values computed by LINflow (in blue) and FastANI (in red) on the *Y* axis for ANI results obtained with pyani (*X* axis) for all pairwise genome comparisons in datasets A through D. Pearson correlation coefficients for FastANI vs. pyani and LINflow vs. pyani results and all other tool comparison results are shown in Fig. 4. (B) Plot showing the differences between the ANI values computed by FastANI and LINflow compared to the pyani ANI values for all pairwise ANI values for datasets A though D.               

comparisons of FastANI (*Jain et al., 2018c*), sourmash (*Pierce et al., 2019*) and pyani (*Pritchard, 2014*) focused on accuracy in assigning strains to species around the 95% ANI species threshold. Here we performed a species level comparison and found LINflow to perform similarly well to sourmash, FastANI and pyani (Fig. 3). However, we then went on and compared the relative performance of all four tools in creating complete similarity matrices from around 70% ANI to almost 100% ANI. We did this comparison using the Mantel test (*Mantel, 1967*) and computing Pearson correlation coefficients for all pairwise tool comparisons, similar to what was done by *Jain et al. (2018c)*, when comparing ANI values obtained by FastANI with ANI values obtained by BLAST. LINflow had the better correlation with pyani compared to sourmash and fastANI except for dataset B, which is composed of highly similar genomes. Even changing the LIN scheme to the 300,000-position LIN scheme did not improve performance of LINflow showing that this was independent of the resolution of the deployed LIN scheme. Moreover, for pairwise ANI values below 85%, we noticed how ANI values computed by LINflow for many different pairs of genomes merged into a few identical ANI values. This is a direct result of the way LINflow infers ANI values based on sequentially assigned LINs. For example, if a hypothetical group of genomes (group 1) contains genomes that are all over 99%

similar to each other and all genomes in a second group of genomes (group 2) are all over 99% similar to each other but the first genome in group 1 that was assigned a LIN and the first genome in group 2 that was assigned a LIN have a pairwise ANI value of only 70.75%, then all pairwise ANI values between group 1 and group 2 genomes will be inferred to be 70.75% by LINflow. Because LINs are assigned sequentially, this is an inherent limitation of LINflow that users need to weigh against the time savings LINflow provides compared to pyani or FastANI.

The comparison between tools also revealed that FastANI's performance is affected by genomic diversity of the analyzed datasets. In fact, FastANI's correlation with pyani was relatively low for datasets A and C, where for each dataset, the genomes are from different species; FastANI's correlation with pyani was high for datasets B and D, where most of the genomes for each dataset are from the same species.

Finally, LINflow stores data in a MySQL relational database that organizes genomic data and the corresponding metadata. MySQL is a relational database that can be accessed from its command-line interface, various application programming interfaces, and graphical user interfaces. Therefore, users can easily retrieve genome sequences for other analyses, for example, comparative genomics or customized reference databases, by querying the database with filters of taxonomic information and/or LINs.

## CONCLUSIONS

LINflow is a fast and memory-efficient pipeline to infer similarity among a large set of prokaryotic genomes and achieves accuracy that approximates, but does not reach, that of pyani. Its ability to quickly add new genome sequences to an already computed similarity matrix makes LINflow particularly useful for projects when new genome sequences need to be regularly added to an existing dataset. Further improvements to LINflow in regard to speed and resolution are underway.

## ACKNOWLEDGEMENTS

The authors acknowledge Advanced Research Computing (ARC) at Virginia Tech for providing computational resources and technical support that have contributed to the results reported within this paper.

### Funding
This study was supported by the National Science Foundation (IOS-1354215) and the College of Agriculture and Life Sciences at Virginia Polytechnic Institute and State University. Funding to Boris A. Vinatzer was also provided in part by the Virginia Agricultural Experiment Station and the Hatch Program of the National Institute of Food and Agriculture, US Department of Agriculture. There was no additional external funding received for this study. The funders had no role in study design, data collection and analysis, decision to publish, or preparation of the manuscript.

## Grant Disclosures

The following grant information was disclosed by the authors:

National Science Foundation: IOS-1354215.

College of Agriculture and Life Sciences at Virginia Polytechnic Institute and State University.

Virginia Agricultural Experiment Station and the Hatch Program of the National Institute of Food and Agriculture, US Department of Agriculture.

## Competing Interests

Life Identification Number and LIN are registered trademarks of This Genomic Life Inc. Lenwood S Heath and Boris A Vinatzer report in accordance with Virginia Tech policies and procedures and their ethical obligation as researchers, that they have a financial interest in This Genomic Life Inc that may be affected by the research reported in this manuscript. They have disclosed those interests fully to Virginia Tech, and they have in place an approved plan for managing any potential conflicts arising from this relationship.

## Author Contributions

- Long Tian conceived and designed the experiments, performed the experiments, analyzed the data, prepared figures and/or tables, authored or reviewed drafts of the paper, and approved the final draft.
- Reza Mazloom conceived and designed the experiments, performed the experiments, analyzed the data, prepared figures and/or tables, authored or reviewed drafts of the paper, and approved the final draft.
- Lenwood S. Heath conceived and designed the experiments, authored or reviewed drafts of the paper, and approved the final draft.
- Boris A. Vinatzer conceived and designed the experiments, authored or reviewed drafts of the paper, and approved the final draft.

## Data Availability

Data is available at Virginia Tech: https://code.vt.edu/linbaseproject/LINflow/

## Supplemental Information

Supplemental information for this article can be found online at http://dx.doi.org/10.7717/peerj.10906#supplemental-information.

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
