# Peer review of "LINflow: a computational pipeline that combines an alignment-free with an alignment-based method to accelerate generation of similarity matrices for prokaryotic genomes"

_PeerJ, doi:10.7717/peerj.10906_

## Round 0.1 · original submission · Major Revisions

Your manuscript has been reviewed by three experts in the field. As you can see from their comments below, all of them raise substantial criticisms on it. Two of them point out that they could not test the software. Please read their comments carefully and revise the manuscript accordingly.

·

Basic reporting

Overall, LINflow seems like an important contribution for accurate and fast ANI calcuations. The workflow seems well thought out and the comparisons to other software seem appropriate.

Experimental design

The approach of the pipeline appears to be robust. When I tried to install the software, I ran into mysql connection issues using 2 different operating systems. I would encourage the authors to try to install LINflow on other systems and make sure that the documentation is complete for installation by basic users on operating systems where they do not have "sudo" privileges. When I look at the code, I notice that the python version is hard coded in the header line. I would encourage the authors to use "/usr/bin/env python" to be consistent with the version that the user has installed through conda. I notice that in LINdb.py, there is some description of creating login files, but this should be added to the main page with enough information to make this accessible to new users.

Validity of the findings

The comparisons to other software seem appropriate. The methods for how the heatmaps were generated should be added to the text. If the hierarchical clustering was performed outside of LINflow, then these methods should also be added to the text.

Additional comments

Based on my issues described above, I wasn't able to test the software, which I would like to do prior to providing more in depth comments. The authors use the term "prokaryote" throughout, although this term is phylogenetically inconsistent. At one point the authors use "microbial". If this term can replace "prokaryote", I would encourage them to change throughout.

In Table 3, the time under LINflow should be in "m" and not "n"

·

Basic reporting

Tian et al present here LINflow, an automated pipeline extending on their previous work with LINbase. The manuscript is generally very clear and well written, and presents a good overview of the relevant literature. I only identified a minor typo in table 3: n instead of m in the LINflow time for set A.

I consider this manuscript and the associated Software an important contribution to the community. However, I remain unconvinced by some assessments, principally due to missing information in the manuscript.

Experimental design

Some design decisions in the Software seem unjustified or only implicitly justified (e.g., k-mer selection in sourmash, or sourmash preference over FastANI). However, this doesn't in any way hinder the reproducibility of the results nor does it affect their validity, and therefore I only raise it as a point of consideration and not really an issue.

More importantly, some key details from the experimental design are not made explicit here. For example, the exact versions of the Software utilized here should be reported (e.g., L196). FastANI recently corrected an important issue in v1.31, and it’d be important to know if the tests were performed before or after this update. Similarly, the method of hierarchical clustering (a central piece on the evaluation of accuracy) is not reported.

Validity of the findings

In Table 3, the authors report a mix of parallelized and linear times, which are not directly comparable. While multi-threaded execution might be the only way to feasibly obtain matrices from pyani, that is not true for FastANI. Why not doing 1 thread for comparison? Note that sub-lineal parallelization as well as computer-specific issues could result in extremely inflated values here. Importantly, I executed FastANI for data set A in a single thread in a laptop and got a result in only 33 minutes (not >11h). While some machine-specific differences are expected, this is a 20-fold difference worthy of reevaluation.

In L200, the authors state: "hierarchical clustering was performed", without specifying the method used or its implementation. The method of hierarchical clustering is a fundamental piece of information, not least due to the fact that none of the dissimilarity metrics used here are metric distances (they all violate the triangle inequality). Note that the only basis of the accuracy evaluation in the manuscript is the cophenetic correlation between dendrograms, not a comparison between distances.

Related to this last point, I (and expectedly other readers) would be curious to know why the authors decided not to evaluate the distances directly. If this were because capturing the "structure" of the distance data is more important than reconstructing actual distances, then a comparison to proper phylogenetic reconstructions (and perhaps different methods of hierarchical clustering) would be warranted. Moreover, direct comparisons of the distances would still be valuable even in this framework. For example, if the actual values are less important than linearity, linear correlations could be evaluated (instead of error). Or, if the monotonicity of distances is more important than linearity, rank correlations could be studied instead. Importantly, the authors indicate there is a good correlation in the ANI range of 70% to 99.8% (L292), but this is never demonstrated.

Finally, the authors present a hypothesis on why the evaluated accuracy of LINflow was low for set B in L283-284. Why not evaluating this hypothesis directly? It seems the authors could simple execute LINflow with the proposed settings and demonstrate the improvement.

Additional comments

I was unsuccessful on deploying the Software presented here. In particular, it appears the installation instructions are incomplete, and indicate a shifting set of prerequisites that has not been fully updated.

The main point of difficulty in the installation was the prerequisites related to MySQL. The manuscript indicates (and justifies) the preference over SQLite in L124, which I personally celebrate. However, the actual code depends upon a "Python MySQL connector" for which no links or installation instructions are provided. It is unclear why is this even necessary (since the databases are reportedly file-based SQLite databases), but if this indeed is necessary additional instructions are required. Note that the advantage of SQLite is that server-based databases are not required. If this is a requirement anyway, there is no advantage on using SQLite. Notably, the conda instructions didn't resolve this dependency either.

Signed: Luis M. Rodriguez-R

·

Basic reporting

1) Figure 3-5 can be combined into one figure. Among them, in Figure 4A, some diagonal ANI values are not highest, and I cannot understand why. Possibly, the order of the genomes in rows and columns are different. For whatever reason, this should be explained.

Experimental design

1) The authors compared LINflow with three other tools, pyani, sourmash, and FastANI, but whereas the purpose of these tools is to calculate ANI between two genomes, the purpose of LINflow is to assign LINs to input genomes. Therefore, direct comparison between these tools is meaningless. The comparison conducted here should be the results of LINflow with other hierarchical classification schemes based on all-against-all ANI calculation by using these programs.
2) In this respect, I was wondering why the authors compared similarity matrices shown in Figures 3-5. How LINflow creates such a similarity matrix while it does not calculate all-against-all ANI? And what is the purpose of this comparison?
3) Although LINflow can calculate the accurate ANI between the query and the most similar genome using pyani, selecting the most similar genome using sourmash may not be so accurate. Therefore, one of the important issues here should be how this inaccuracy affects the final result. However, the experimental tests in this manuscript seem not to address this issue adequately.

Validity of the findings

1) The authors insist that LINflow is more accurate than FastANI for the dataset A and C in terms of the cophenetic correlations shown in Figure 6 and this is because these datasets contain different species (lines 285-289). However, in these cases, sourmash is also better (more similar to pyani) than FastANI. This seems to contradict the previous observation by the authors of FastANI (Jain et al. Nat Comm. 2018), where the accuracies of FastANI is better than Mash especially when more divergent genomes are compared. The authors should explain this point more clearly.
2) Similarly, I cannot find any clear evidence for the following sentence (lines 291-292): LINflow reveals genomic relationships more accurately than FastANI and sourmash over the range of ANI values from 70% to around 99.8%.

Additional comments

LINflow is a system for assigning a LIN (Life Identification Number) to each genome in a set of genome sequences. The resulting set of LINs corresponds to a hierarchical classification of the input sequences based on ANI. LINflow avoids calculating all-against-all ANIs to calculate LINs by incremental updating process, and utilizes two existing programs, sourmash and pyani, for rapidly identifying the most similar genome in the set and for rigorously calculating the ANI between the query and the most similar genome, respectively. I think that this scheme is reasonable enough, and possibly useful for maintaining a large-scale comparative genome database.
As I mentioned in the Experimental design section above, I do not think that direct comparison between LINflow and other ANI calculation tool is meaningful. In fact, I think that ANI calculation tools used in LINflow can be replaced with other tools such as FastANI, and possibly it may improve the performance. Therefore, I recommend the authors not to focus on comparison of LINflow with other ANI calculation tools, but more focus on designing an efficient genomic data classification scheme.

---

## Round 0.2 · Major Revisions

Your revised manuscript has been reviewed by the same three original reviewers. As you can see from their comments below, one of them now recommends its acceptance (with a very minor revision) while the other two still are not satisfied with your responses well. Please read their comments carefully and re-revise the manuscript accordingly (if you agree with them). Looking forward to your re-revised manuscript.

·

Basic reporting

no comment

Experimental design

no comment

Validity of the findings

no comment

Additional comments

The authors obviously made an attempt to improve both the manuscript and the software package in this resubmission. Unfortunately, I follow the instructions on the github page for conda and am still unable to run the program on centOS, ubuntu, or MacOSX. The problem is still mysql. I could dig into getting this fixed, but seems beyond the scope of this particular review.

My cluster is unable to run docker images, so I wasn't able to test the docker image on centOS. I also tried on ubuntu, but looks like I would need sudo privileges to set up the container, which I'm unable to do.

Unless the software is easy to install, it is unlikely to be widely used. I would encourage the authors to rethink their dependencies or improve the documentation to allow for each installation and usage in the absence of sudo privileges. One way to test this is to have users on different machines work through the instructions and verify that they can install. Additionally, it appears that the authors didn't change the first line to "#/usr/bin/env python", which will cause serious conflicts if a user tries to invoke the script without a preceding "python".

·

Basic reporting

I don't have any substantive comments left. I congratulate the authors on this significantly improved manuscript, that I'm sure will have an important impact in the field. I'm eager to use LINflow in production myself.

Only one (very) minor comment: the new text introduces the capitalization pyANI for the tool referred elsewhere as pyani. I believe the all-lowercase version is the correct styling.

Experimental design

No additional comments.

Validity of the findings

No additional comments.

Additional comments

No additional comments.

Signed: Luis M. Rodriguez-R

·

Basic reporting

My second question has not been addressed yet. My question is: if rows and columns of the matrix are in the same order, the diagonal should correspond to the ANIs of the same strains, which should be 100. But this seems not to be true for FastANI (Fig.3B). Is it correct?

In addition, although the legend of Fig.3 says that there is a supllmenentary figure showing the same matrix with strain names (which can be used to confirm the order of the matrix), I cannot find any such supplementary figure.

Experimental design

I still feel that the authors’ assertion that the main purpose of LINflow is to calculate a similarity matrix is somewhat strange. Instead, I think that the main purpose should be classifying genomes (with accurate ANI values) for which a similarity matrix is required. And this is why LINflow can be a useful tool even if it does not calculate all-against-all similarities assuming that genomic data can be classified in a hierarchical manner. Although I do not force to do so, I recommend the authors to mention this point.

Validity of the findings

No comment

---

## Round 0.3 · accepted · Accept

Your revised manuscript has been reviewed by two of the original reviewers, who raised some points. Now, since both of them recommend its acceptance, I am happy to recommend its acceptance.

Congratulations!

·

Basic reporting

No comment

Experimental design

no comment

Validity of the findings

no comment

Additional comments

I'm still having difficulties getting LINflow installed, but perhaps I'm an anomaly and am willing to accept the manuscript and let the community decide on the usability of the software.